# Light Intensity Affects the Assimilation Rate and Carbohydrates Partitioning in Spinach Grown in a Controlled Environment

**DOI:** 10.3390/plants12040804

**Published:** 2023-02-10

**Authors:** Simona Proietti, Roberta Paradiso, Stefano Moscatello, Francesco Saccardo, Alberto Battistelli

**Affiliations:** 1Research Institute on Terrestrial Ecosystems (IRET), National Research Council of Italy (CNR), 05010 Porano, Italy; 2Department of Agriculture, University of Naples Federico II, 80055 Naples, Italy; 3Department of Agriculture and Forest Sciences (DAFNE), University of Tuscia, 01100 Viterbo, Italy

**Keywords:** *Spinacia oleracea* L., photosynthesis, source-sink relationship, sucrose, starch, enzyme activity, FBPase, SPS, AGPase

## Abstract

The cultivation of spinach (*Spinacia oleracea* L.) has been increasing during the last years in controlled environment agriculture, where light represents a key factor for controlling plant growth and development and the highest energetic costs. The aim of the experiment was to evaluate the plant’s response to two light intensities, corresponding to an optimal and a reduced level, in terms of the photosynthetic process, photoassimilates partitioning, and the biosynthesis of sucrose and starch. Plants of spinach cv. ‘Gigante d’Inverno’ were grown in a phytotron under controlled conditions, comparing two values of photosynthetic photon flux density (PPFD), 800 μmol m^−2^ s^−1^ (800 PPFD) and 200 μmol m^−2^ s^−1^ (200 PPFD), at a 10 h light/14 h dark regime. Compared to 800 PPFD, under 200 PPFD, plants showed a reduction in biomass accumulation and a redirection of photoassimilates to leaves, determining a leaf expansion to optimize the light interception, without changes in the photosynthetic process. A shift in carbon partitioning favouring the synthesis of starch, causing an increase in the starch/sucrose ratio at the end of light period, occurred in low-light leaves. The activity of enzymes cFBAse, SPS, and AGPase, involved in the synthesis of sucrose and starch in leaves, decreased under lower light intensity, explaining the rate of accumulation of photoassimilates.

## 1. Introduction

Plant production in controlled environment agriculture (CEA) is proposed as the modern evolution of agriculture. CEA is an advanced farming technology under controlled conditions, in terms of light, temperature, carbon dioxide, relative humidity, and nutrient supply, and also in presence of an unfavourable outdoor climate and at high cultivation density [1]. It aims to guarantee high and constant year-round production, food quality and security, resilience to climate change, and sustainability, and allows high resource use efficiency and better pathogen control. In addition, CEA has the potential to meet the increasing food demand of the world population, and some applications, such as vertical farms (VF), also to shift the food production closer to urban centres.

Light influences the plant growth, development, and metabolisms, hence the yield and nutritional quality of products. Specifically, light quantity (intensity and duration) and quality (wavelength composition) affect plant behaviour throughout the entire life cycle, as plants use light as both the energy source for photosynthesis, and as a signal to activate and regulate many other fundamental processes in photomorphogenesis [2]. In absence of other limiting factors, an increase in the light intensity, as photosynthetic photon flux density (PPFD), will result in an increase of the photosynthetic rate and, likewise, of the plant yield. Hence, artificial lighting is a key factor to modulate the plant performance in protected cultivation, and the fine control of light is still one of the major technical challenges to develop successful CEA systems and, more specifically, vertical farms. Vertical farming is a fast-growing sector in agriculture; however, it implies highly innovative and technological content, among which lighting represents the most critical point and the highest cost [3]. As a consequence, light use efficiency is crucial in these systems for the plant performances and the environmental and economic sustainability of the production process [4].

Light intensity influences the plant source-sink relationships. In general, the produced photosynthates are translocated from source organs (e.g., mature leaves) to sink organs (e.g., fruits, new shoots, and roots), promoting the biomass accumulation and the growth of sink tissues, thereby significantly influencing the crop yield and quality [5]. The sink strength generally depends on the size (total weight) and activity (rate of uptake of transport sugars per unit weight) of sink organs, and the alteration of both these parameters can modulate the translocation patterns of photoassimilates [6,7].

The carbohydrates synthesis at the source level and their utilization at the sink level (metabolisms of sink organs) represent two processes strongly related that directly affect the photosynthetic rate. Each event (including light deficiency) limits the sink activity or the sucrose transport determines a sucrose accumulation at the source level, increasing the expression of genes involved in carbohydrate storage, and repressing that of photosynthetic genes, finally reducing photosynthetic capacity. On the contrary, the high activity of sink organs requests a high level of photoassimilates to sustain the growth and metabolic processes, and maintains a high photosynthetic rate [8,9].

The partitioning of photoassimilates occurs at the level of a leaf’s individual cells, where two main non-structural carbohydrate pools, namely the sucrose and the starch pools, are formed [10]. The partitioning in the two streams takes place in the cytoplasm [11], where the carbon assimilated in the triosophosphates (TP) can be converted in the cytosol into sucrose, then exported to the rest of the plant, or stored in the chloroplast for the synthesis of starch [10,12]. In order to sustain the plant development, the pools of sucrose and starch in the leaves are subjected to a high turnover [13], regulated by biochemical processes of synthesis and degradation in the short term [12,14], and by changes in the gene expression influencing the of the whole plant metabolisms in the long term.

In autotrophic organisms, the fine regulation of sucrose and starch synthesis is controlled by three enzymes: the two cytosolic enzymes fructose-1,6 bisphosphatase (E.C. 3.1.3.24, FBPase) and sucrose phosphate synthase (E.C. 2.4.1.14, SPS), which regulate the sucrose synthesis, and the plastidial ADP-glucose pyrophosphorylase (ATP: α-D-glucose-1 phosphate adenyltransferase, E.C. 2.7.7.27, AGPase), which controls the starch synthesis in leaf chloroplasts and cells of all sink organs accumulating starch.

The export of non-structural carbohydrates from the leaf is regulated by their status, which affects the activity of enzymes involved in the synthesis or degradation of sugars, and the expression of carriers responsible for their transport to the entire plant [15]. In higher plants, about 80% of photoassimilates are transported as sucrose, and both the symplastic transport, via plasmodesmata, and the active apoplastic loading, through specific transporters, cooperate for the sucrose export from the photosynthetic cells to the phloem [16,17].

The cultivation of spinach (*Spinacia oleracea* L.) is widespread in world horticulture. Thanks to the fast growth rate, the small plant size, and the nutritional properties, it is an ideal crop for CEA. Furthermore, spinach is a model plant for plant physiology and biochemistry investigations on photosynthesis, and carbohydrate and nitrogen metabolisms. It is considered a “sucrose forming” species as a specific carrier protein, namely the *So*SUT1 (*Spinacia oleracea* Sucrose Transporter 1), acts as co-transporter of Sucrose/H+ for phloematic loading of sucrose through the apoplast.

In cultivation systems in CEA, a deep understanding of the dynamics of photosynthates translocation depending on the light environment can be relevant to increase the crop’s yield and quality and to ensure highly profitable and stable productions. In our work, spinach plants were grown in pot in a growth chamber, under controlled environment, comparing the optimal light intensity of 800 μmol m^−2^ s^−1^ (800 PPFD) and a sub-optimal light intensity of 200 μmol m^−2^ s^−1^ (200 PPFD), provided in a 10 h light/14 h dark regime. The effects of acclimation to high and low light on the distribution of photoassimilates in the starch and sucrose pools were studied, considering two key mechanisms: the balance between the synthesis and mobilization of sugars, and the regulation of biosynthetic processes of sucrose and starch. The photosynthetic parameters, the non-structural carbohydrates of leaves, and the activity of the major enzymes involved, namely FBPase, SPS, and AGP, were also quantified.

Compared to previous studies on spinach metabolism available in literature, our experiment aimed at acquiring information related to a more advanced growth stage. Indeed, we performed our measurements and analyses on adult plants, as data from most of the other experiments with similar scope seems to refer to younger plants. In our opinion, characterizing the plant response to light stimuli throughout the entire growth cycle is crucial to set up efficient protocols for artificial lighting, based on the real light requirements of plants in the specific growth and developmental stages. This information has relevant value for the potential practical application in vertical farms, in which artificial lighting represents one of the most energy requiring and expensive operations. In this respect, evaluating the plant response to sub-optimal lighting conditions in the different vegetable crops would improve the economical sustainability of vertical farming.

According to the literature on light requirement for photosynthesis in spinach [18,19], the maximum assimilation rate in this crop can be reached at a PPFD around 1000 mmol m^−2^ s^−1^. On this basis, we established our lighting treatments at 200 µmol m^−2^ s^−1^, to simulate a possible sub-optimal light level feasible in sustainable vertical systems (taking into account the high energy cost), and 800 µmol m^−2^ s^−1^, representing the optimal light intensity for the crop (preventing excess light stress and photoinhibition).

## 2. Results

### 2.1. Plant Growth and Yield, and Product Characteristics

Under high light intensity, both fresh and dry weight of leaves and roots significantly increased, and the dry matter percentage was, respectively, 12.6% and 15.7% higher compared to low light intensity (Table 1).

The total leaf chlorophyll content did not show significant difference between the lighting treatments (463.8 ± 31.8 and 501.9 ± 48.6 mg/leaf m^2^ for low and high light, respectively), and the Chl a/b ratio was similar too (3.05 ± 0.05 and 2.91 ± 0.09).

### 2.2. Gas Exchange

Table 2 shows the values of the main photosynthetic parameters recorded in plants grown under the two light intensities, after different times of exposure to lighting. 

Leaves of spinach plants under 200 μmol m^−2^s^−1^ PPFD showed an assimilation rate (A) three times lower compared to those at 800 μmol m^−2^s^−1^ (5.0 ± 0.1 and 15.4 ± 0.4 µmol CO_2_ m^2^s^−1^, respectively) (Table 2). Conversely, the value of intercellular CO_2_ concentration (Ci) was 13.7% higher in leaves under lower irradiance (360.3 ± 0.8 vs. 310.6 ± 2.6 µbar bar^−1^ at high irradiance), while stomatal conductance (g_s_) was unaffected by the light intensity (470.3 ± 15.5 and 517.4 ± 35.3 mmol CO_2_ m^2^ s^−1^ for 200 PPFD and 800 PPFD, respectively) (Table 2).

The interaction between the two factors (light intensity and times of exposure) was significant for the assimilation rate (A) and substomatal CO_2_ partial pressure, while the two factors did not affect stomatal conductance (g_s_) (Table 2). The CO_2_ uptake and Ci value did not change significantly during the light period, while the stomatal conductance (gs) seems to decrease after 8 h of lighting, although this difference did not result statistically significant (Table 2).

The influence of lighting treatment on the relation between CO_2_ assimilation and intercellular CO_2_ concentration (A–Ci) is shown in Figure 1. Plants grown at 800 and 200 PPFD showed a similar trend of the A/Ci, proving that a higher concentration of intercellular CO_2_ is not necessary to support a similar rate of photosynthesis of plants acclimated to 200 μmol m^−2^ s^−1^ compared to those acclimated to 800 μmol m^−2^ s^−1^.

### 2.3. Non-Structural Carbohydrate Content

Spinach plants grown at high light intensity showed a higher content of soluble sugars (glucose, fructose, and sucrose; Figure 2A) and total carbohydrates (soluble plus starch Figure 2B), compared to those at low-light intensity. The difference in carbohydrate content was minimal at the beginning and increased to a maximum at the end of the light period.

Sucrose showed a higher concentration than glucose and fructose, under both high and low light (Figure 3A–D). In particular, in plants under high light the concentration of sucrose was greater by about 20 times than that of glucose (Figure 3A,C), and up to 30 times than that of fructose (Figure 3C,B). 

Furthermore, in the same plants, the amount of sucrose accumulated in leaves at the end of the light period was about 29% greater respect to that of starch (Figure 3C,D and Table 3). 

Moreover, in plants at high irradiance, the concentration of glucose, fructose, sucrose, and starch increased significantly during the light period, while leaves acclimated to low light accumulated less soluble carbohydrates, and only the starch increased 20 times at the end compared to the beginning of the photoperiod (Figure 3D).

The different carbon partitioning assimilated in leaves grown at low irradiance was underlined by the starch/sucrose ratio, which was similar between the two lighting treatments early in the light period, while it reached a value 60% lower in plants under 800 PPFD compared to those under 200 PPFD at the end of the lighting period (Figure 4).

At the end of the light period, in leaves acclimated at low light, 68.5% of total carbohydrates were partitioned towards the formation of starch reserves, while at high irradiance 43.6% of the sugars consisted of starch. The percentage of soluble carbohydrates was two times lower than that of starch in spinach leaves grown at 200 PPFD, while soluble carbohydrates were higher than starch at 800 PPFD (Table 3).

In plants grown at 200 PPFD, the total content of carbohydrates at the beginning of the light period, after the dark phase, was about 4 times lower compared to that at high light, reflecting the decreased availability of assimilates, due to the reduction of carbon assimilation. At the beginning of a new light period, the percentage of starch reached about 30% of total assimilates for both low-light and high-light spinach leaves (Table 3).

The amount of the assimilated carbon exported from the chloroplast and accumulated as non-structural carbohydrates, in relation to the carbon assimilation rate, is shown in Table 4. 

In spinach leaves under low and high light intensity, 390 and 1050 µmol C mg^−1^ chl were fixed, respectively, during the 10 h light period, considering a constant photosynthesis rate of 39 and 105 µmol CO_2_ mg^−1^ chl h^−1^, respectively. In leaves under 200 and 800 PPFD, 37.4% and 38% of the total carbon assimilated was allocated during the day as non-structural carbohydrates, respectively.

The assimilates exported during the day, calculated as the difference between the fixed CO_2_ and the total carbohydrate accumulated and reported in Table 3, showed a value 2.7 times lower in leaves at 200 PPFD than 800 PPFD. The rate of assimilation during the day and night was lower for low-light leaves, and the translocation of assimilates during the night amounted to 43% and 44% of the translocation rate during the day, for low and high light leaves, respectively (Table 4).

### 2.4. cFBPase, SPS, and AGPase Activity

Both the maximum SPS activity (SPS_max_) and AGP activity were significantly affected by light intensity (Table 5). 

In spinach grown in low light conditions, SPS_max_ and AGPase showed 36% and 20% lower values than in high light (Table 5), respectively, while cFBPase was not significantly different between the two treatments. During the light period, the activity of cFBPase and SPS_max_ did not change significantly, while that of AGPase was 23.7% higher in the morning than in the afternoon (Table 5). The activation state of SPS was three times higher in leaves grown in high light than in those in low light (respectively 33% and 13%). In high light growth conditions, the activation state of SPS halved its value from 4 to 8 h of light exposure (41.5% to 24.5%), while under 200 PPFD it was always lower (15.9% and 10.5% at 4 and 8 h, respectively), with no significant differences between the hours of the day. In leaves grown at 200 μmol m^−2^ s^−1^, the AGPase/SPS_max_ was constant in the light period, while in high light it decreased sharply, in accordance with the trend of starch/sucrose ratio (Figure 5).

## 3. Discussion

### 3.1. Plant Growth and Yield, and Product Characteristics and Gas Exchange

Plants can acclimate to changes in light intensity by adapting the leaf anatomical structure, plant morphology, and photosynthetic parameters (rate of assimilation, transpiration, and stomatal conductance), which affect the biomass accumulation [20]. In our experiment, low light intensity significantly reduced the CO_2_ assimilation of spinach plants (A), without changes in stomatal conductance (g_s_). This evidence, together with the higher value of Ci in plants under low light, indicated that the reduced assimilation cannot be ascribed to stomatal limitations of the photosynthetic process [21].

The A/Ci relation in plants under both the light conditions showed a similar trend, suggesting that spinach leaves exposed to different irradiance did not differ substantially in the Rubisco activity, as demonstrated by the linear portion of the response curve. Similarly, no difference was observed in the efficiency of electron transport and the regeneration of Ribulose 1,5 bisphosphate (RuBP), as defined by the final part of the curve [22]. This result agrees with the evidence that Rubisco was in excess in low light plants [23] and that this protein does not exert a control on the photosynthetic rate for a wide range of growth light intensities [24]. Moreover, the chlorophyll content and the Chl a/b ratio did not differ in spinach leaves under the two light regimes, indicating that the photosynthetic apparatus and its functionality were unaffected by light intensity [25]. In coffee seedlings, Rodríguez-López et al. [26] stated that biomass accumulation increased linearly with increasing light intensity, and Wu et al. [27] showed that in soybean, changes in light intensity led to considerable differences in leaf morphology and structure. In a previous experiment [28], we demonstrated that low light affected the growth of spinach, reducing the leaf thickness and plant production and increasing the shoot/root ratio. In this experiment, our data showed that in low light conditions, the leaf dry weight per plant was five times higher than the root dry weight, while this ratio was only three times in spinach plants grown in high light conditions, indicating a change in carbon allocation on the epigeal organs, as a consequence of the leaf expansion promoted to optimize the light interception.

### 3.2. Accumulation of Assimilates in Leaves and Translocation Rate

Plants can adjust the rate of sucrose export, as well as of starch biosynthesis and degradation to the light availability [29], and it is well known that starch formation increases in plant grown under high light intensity, due to enhanced carbon assimilation and photosynthetic product availability [30].

Sucrose is the main soluble sugar in leaves of spinach, which can be considered a sucrose accumulator species. Our data highlights this feature, as at high light plants showed a soluble carbohydrate content about 1.6 times higher than starch. However, leaves acclimated to different light conditions by modifying the partitioning of assimilated carbon during the day, as demonstrated by the daily variation of photoassimilates. Under low light intensity, in particular late in the light period, spinach leaves directed the assimilated carbon towards the starch synthesis rather than in soluble carbohydrates (68.1% vs. 31.8%), represented mainly by sucrose, in an opposite trend compared to leaves acclimated to high light (43.6% vs. 56.3%).

The leaf carbohydrate content in spinach decreased substantially during the night in both plants at 200 and 800 PPFD. Studies on source–sink relationship in *Arabidopsis thaliana* at the vegetative phase showed a similar trend of sugar content, with a progressive drop of carbohydrate amount, particularly during the night, related to the stunting growth in the dark and the depletion of starch reserves [31]. Spinach leaves considered in the present work were fully expanded source organs, so the starch reserves were essentially used to maintain their metabolism, and at the beginning of a new light period, low light leaves had mobilized the 51% of the accumulated starch, while high light leaves, only the 34%. It is well known that a decrease in carbohydrates availability can enhance the expression of the triose-phosphate translocators (TPT) gene, related to photosynthetic systems and to the mobilization and export of reserves, as starch that could buffer the reserve shortage [32,33].

The net assimilation rate was different for low and high light spinach leaves, so the production of photoassimilates during the 10 h of light exposure was 67% lower in leaves at 200 PPFD than at 800 PPFD. Considering these values, the total accumulation was different under low or high light intensity, resulting in a 2.7 times lower value in leaves at 200 PPFD, but in a similar percentage of total assimilates compared to 800 PPFD (37.4% and 38.0%, respectively). The average rate of assimilates utilization during the night period was lower than in the light period, for both low and high light intensity, as reported by several authors under different conditions [32,34], although the night translocation was similar, amounting to 43 and 44% under 200 and 800 PPFD, respectively, of the daily rate.

The leaf accumulation of assimilates in leaves and the translocation rate were evaluated without a direct quantification of photorespiration. However, during the light period, photorespiration was already taken into account, as the net photosynthesis corresponds to CO_2_ fixation minus CO_2_ release. During the dark period, the CO_2_ released through the respiration process was quantified to about 2% of the assimilate utilization [32], hence, we can assume that this process has only a negligible impact on the assimilate utilization.

### 3.3. Activity of cFBPase, SPS, and AGP

The modulation of the carbohydrate partitioning between sucrose and starch was linked to changes in the activity of the key enzymes, cFBPase, SPS, and AGPase, that have a high control coefficient in sucrose synthesis in the cytosol (cFBPase and SPS) and starch synthesis in the chloroplast (AGPase) [10]. Reduced carbon produced by the Calvin cycle in the chloroplast is exported to the cytosol as triose phosphates and channeled to sucrose synthesis by the coordinate activity of several enzymes, with cFBPase and SPS playing a special role [35]. Alternatively, assimilates are retained in the chloroplast and used for primary starch synthesis. The flux through the two alternative photosynthetic products synthesis (sucrose and starch) are regulated by several variables, but a relevant role is played by metabolic factors (e.g., allosteric modulators), and by the genetic control of the enzyme maximum activity [35,36,37]. We have tested changes in the maximum activity of all three enzymes and the activation state of SPS [36].

The balance between starch and sucrose increases when cFBPase activity is reduced [36]. In the present work, a tendency to a reduction of cFBPase activity was observed in leaves at 200 PPFD, although the difference from leaves at 800 PPFD was not statistically significant. Strand and co-authors [38] showed that a decreased expression of cFBPase determined a switch from sucrose to starch synthesis, together with changes in metabolites relationship, such as a five to sixfold higher PGA:triose-P ratio, decreased Ru1,5bisP, and inhibition of photosynthesis. Preliminary works showed a decreased photosynthetic capacity in spinach grown in similar conditions, but at a younger growth stage, indicating that light intensity can cause a joint acclimation of photosynthesis and photosynthetic end-product synthesis capacity and fluxes [39]. In the conditions of our experiment, no sign of a decreased photosynthetic capacity was measured (similar A vs. Ci partial pressure, no change in chlorophyll content), even in the presence of a relatively higher starch and sucrose synthesis in late light time under 200 PPFD. This discrepancy is likely due to the longer acclimation period in our plants, which were older than in Battistelli et al. [39]). This evidence demonstrates the high versatility of plant acclimatory responses to light and, even more importantly, that the acclimation of starch and sucrose biosynthetic pathways in the photosynthetic cells can occur even in the absence of an evident modification of photosynthetic capacity. This point indicates that balancing photoassimilates partitioning at the photosynthetic level is of paramount importance and is a very sensitive phenomenon in response to the prevailing light environment. It also demonstrates that an efficient acclimation of photosynthate partitioning can avoid feedback inhibition of photosynthesis due to the lack of Pi recycling at the cellular level [40].

In conclusion, our work shows that in spinach leaves the acclimation to low light intensity was reached through modifications of the biomass allocation and of carbon distribution at the plant level. Spinach plants grown under low irradiance reduced biomass production and redirected the photoassimilates to leaves to optimize light interception. This change occurred without effects in the functionality of the photosynthetic process, and the reduction of photosynthetic rate was due essentially to the low radiative energy available for low light leaves. Moreover, a shift in the partitioning of carbon assimilated favouring starch synthesis was observed in leaves under low light inducing an increase in starch/sucrose ratio. Besides, the estimated synthesis and export rates of assimilates showed that the translocation during the night amounted to 43% and 44% of the rate observed in the day, for low and high light, respectively. Enzymes involved in the synthesis of sucrose and starch such as cFBAse, SPS, and AGPase lowered their activity, explaining the rate of synthesis and accumulation of photoassimilates. In terms of yield and quality, the lower light intensity decreased the crop production of spinach in a controlled environment.

## 4. Materials and Methods

### 4.1. Plant Material and Growth Conditions

Spinach seeds (*Spinacia oleracea* L. ‘Gigante d’Inverno’), from Hortus Sementi (Longiano, Forlì-Cesena, Italy), were sown in perlite in 1 l plastic pots (three seeds per pot), placed in a growth chamber (Fitotron SGD170 Sanyo Gallenkamp, Cambridge, UK) provided with a 10 h light/14 h dark photoperiod, 200 µmol m^−2^ s^−1^ photosynthetic photon flux density (PPFD, l 400–700 nm) of white light, at 360 ppm CO_2_. Temperature and relative humidity were 25 ± 0.5 °C/70% and 20 ± 0.5 °C/70% during the light and the dark periods, respectively. After the expansion of cotyledons, pots were placed at 800 µmol m^−2^ s^−1^ and 200 µmol m^−2^ s^−1^ for the optimal and the sub-optimal light treatment, respectively. The light was provided by white light lamps (Osram Powerstar HQI^®^-TS 250W/NDL UVS).

Plants were supplied twice per week with a full-strength Hoagland nutrient solution [41], with a pH of 6.5 and EC 1.7 mS/cm, and with fresh water.

### 4.2. Sampling and Analysis

Five weeks after the expansion of cotyledons, 12 plants from each treatment were randomly collected, shoots and roots were separated, and fresh weight (FW) was measured. Dry weight (DW) was measured after oven drying at 80 °C. The dry matter percentage (DM%) was calculated as DW/FW × 100. The 4th and 5th fully expanded leaf was used to determine the content of non-structural carbohydrates (glucose, fructose, sucrose, and starch) and enzymatic activity. For non-structural carbohydrate (NSC) determination, leaf samples were collected at the end of the dark period and after 4 h, 8 h, and 10 h of light, while for enzymatic activity measurements, after 4 h and 8 h of light.

### 4.3. Gas Exchange

Gas exchange was measured in an open system using a Cira 1 PP System (Portable Photosynthesis System, Hitchin, UK) as in Ripullone et al. [42]. Assimilation rate (A), stomatal conductance (gs), and the internal CO_2_ concentration (Ci) were measured in spinach leaves of plants grown at 200 and 800 µmol m^−2^ s^−1^ after 4 h, 6 h, and 8 h of light exposure. Measurements of gas exchange were carried out at the environmental parameters previously described. The A/Ci curves were obtained by setting the CO_2_ concentrations at 400, 200, 50, 70, 90, 110, 120, 180, 200, 250, 290, 300, 410, 550, 800, 850, 1000, and 1200 ppm, under the saturating light intensity of 1200 μmol m^−2^ s^−1^. At each point, data were recorded when the intercellular CO_2_ concentration reached a stable value. Measurements were conducted at a leaf temperature of 25 °C, with a flow rate of 500 µmol s^−1^.

### 4.4. Carbohydrate Content

Measurements of non-structural carbohydrate (NSC) were performed on 1 cm^2^ leaf discs, collected from leaf lamina with a cork borer, and immediately frozen in liquid nitrogen. Soluble carbohydrates were extracted at 80 °C for 45 min, after grinding the leaf discs in a glass-glass homogeniser, containing 1.5 mL of 80% (*v*/*v*) ethanol, 20% (*v*/*v*) 100 mM Hepes (pH 7.1), and 10 mM MgCl_2_. After centrifugation at 16,000× g for 5 min, the supernatant was analysed for soluble carbohydrate content. Glucose, fructose, and sucrose were determined by an enzymatic coupled spectrophotometric assay with hexokinase (1.2 U), glucose phosphate dehydrogenase (0.3 U), phosphoglucose isomerase (0.3 U), and acid invertase (30 U) as described in Moscatello et al. [43].

Starch was quantified from the pellet remaining after extraction of soluble sugars, after three washing with 50 mM NaAcetate buffer (pH 4.5), suspended in 1 mL of the same buffer, and autoclaved at 120 °C for 45 min. The suspension was incubated at 50 °C for 1 h with amyloglucosidase (40 U) and α-amylase (4 U). The glucose produced by starch hydrolysis was determined enzymatically as described above.

### 4.5. Enzymatic Activity (cFBPase, SPS, AGP)

The measurement of cytosolic fructose-1,6 biphosphatase (E.C. 3.1.3.24, cFBPase) activity was performed according to the method of Holaday et al. [44]. Spinach leaves were extracted by grinding 80 mg of tissue in a glass-glass homogenizer, containing 1 mL of ice-cold buffer containing 50 mM Hepes pH 7.0, 5 mM MgCl_2_ 1 mM EDTA, 10 mM DTT, 0.2% (*v*/*v*) Triton X100, and 0.2% (*w*/*v*) PVPP. Each extract was centrifuged at 16,000× g for 4 min at 4 °C and the supernatant was used for the determination of the enzymatic activity of cFBPase. Leaf extracts, used for enzymatic assays of sucrose phosphate synthase (E.C. 2.4.1.14, SPS) and ADP-glucose pyrophosphorylase (E.C. 2.7.7.27, AGP), were prepared by homogenizing 200 mg spinach leaf in 1.5 mL of an ice-cold buffer consisting of 50 mM Hepes pH 7.4, 5 mM MgCl_2_, 1 mM EDTA, 1mM EGTA, 2.5 mM DTT, and 2 mM Benzamydine, with 10% (*v*/*v*) Glycerol and 0.4% (*v*/*v*) Triton X100. Leaf extracts were centrifuged at 16,000× g for 4 min at 4 °C and the supernatant was desalted through Sephadex G-25 (Pharmacia Biotech, Uppsala, Sweden), with the same buffer used for the extraction but without Triton X100.

The SPS activity was first measured in the presence of saturating concentrations of hexose phosphates to check the enzymatic maximum activity: the extract, containing 4 µg chlorophyll, was incubated for 15 min at 20 °C with 2 mM fructose-6 phosphate (Fru-6P), 10 mM glucose-6-phosphate (Gl-6P), and 3 mM uridine diphosphoglucose (UDPG), 50 mM Hepes (pH 7.4), 4 mM MgCl_2_, and 1 mM EDTA. The SPS activity in presence of limiting conditions was measured as described above but adding 5 mM inorganic phosphate (Pi). The assay solution of 2 mM Fru-6P and 10 mM Glc-6P was used as a control. After incubation, the assay solutions were heated to 95 °C for 3 min. After this treatment, solutions were centrifuged for 2 min and an aliquot was taken to measure the UDP formed, using a dual-wavelength spectrophotometer [36].

The pyrophosphorolytic activity of AGP was assayed spectrophotometrically, measuring the formation of NADH at 340 nm and 25 °C [45]. The assay mixture contained: 100 mM glycylglycine-NaOH (pH 7.5), 10 mM MgCl_2_, 2 mM DTT, 5 mM NaF, 0.5 mM NAD^+^, 1 mM ADPglucose, 100 µM glucose 1,6-bisphosphate, 1 mM tetra-sodium pyrophosphate (PPi), and 2 units each of phosphoglucomutase (from rabbit muscle and glucose 6-phosphate dehydrogenase (from *Leuconostoc mesenteroides*). After correcting for non-specific reduction of NAD^+^ by samples, the reaction was initiated by the addition of PPi.

### 4.6. Statistical Analysis

Statistical analysis was done by ANOVA using the STATISTICA software package (StatSoft for Windows, 1998, Tulsa, OK 74104 United States).

Growth characteristics were analysed by one-way analysis of variance (ANOVA). Data on carbohydrate contents and enzymatic activity were analysed using two-way ANOVA, with the time of sampling and light intensity as factors. Differences between averages were tested by Fisher’s post-hoc test, with a significance level of *p* = 0.05 and designated with different letters.

## 5. Conclusions

We produced new evidence on how controlled growth conditions can affect plant growth, photosynthesis, and carbohydrate amount and partitioned it at both the leaf and plant level. Modifications of photosynthesis and growth in response to light environment are key aspects in indoor plant cultivation, dramatically impacting the energy conversion efficiency, productivity, and produce quality [39]. Furthermore, soluble carbohydrates in leaves are a key determinant of the taste of vegetables, and total carbohydrates at harvest might influence the cell survival during post-harvest handling and commercialization, thereby affecting the product shelf-life and consumer acceptance. Fully controlled environment agriculture can result in energetic and economical sustainability by better knowing the biochemical and physiological determinants of plant response to the growth environment.

## Figures and Tables

**Figure 1 plants-12-00804-f001:**
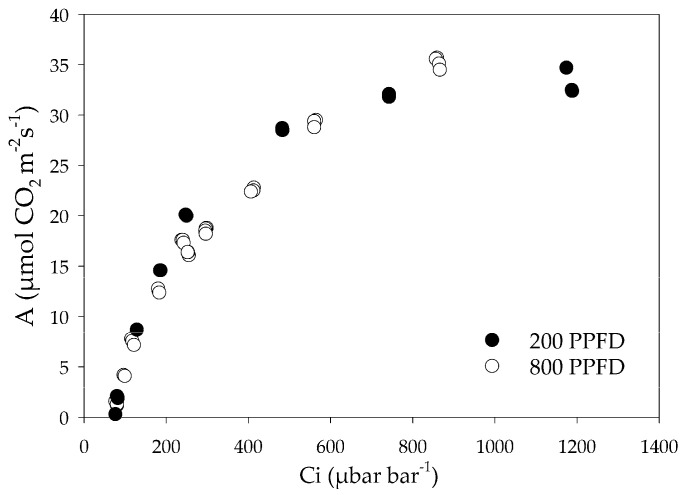
Relation between CO_2_ assimilation and intercellular CO_2_ concentration (Ci) in leaves of spinach grown in a growth chamber under 800 μmol m^−2^ s^−1^ (800 PPFD) and 200 μmol m^−2^ s^−1^ (200 PPFD) white light intensity.

**Figure 2 plants-12-00804-f002:**
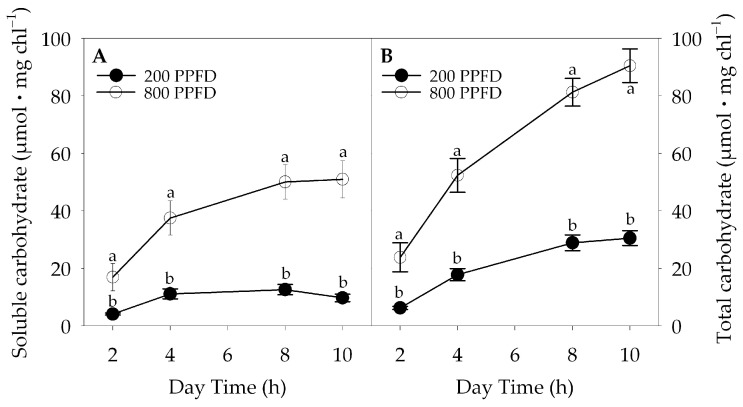
Soluble (**A**) and total carbohydrates content (**B**) during the light period in leaves of plants grown under 800 μmol m^−2^ s^−1^ (800 PPFD) and 200 μmol m^−2^ s^−1^ (200 PPFD) white light intensity. Mean values ± S.E. (n = 12). Different letters indicate significant differences at *p* = 0.05 (Fisher’s post-hoc test).

**Figure 3 plants-12-00804-f003:**
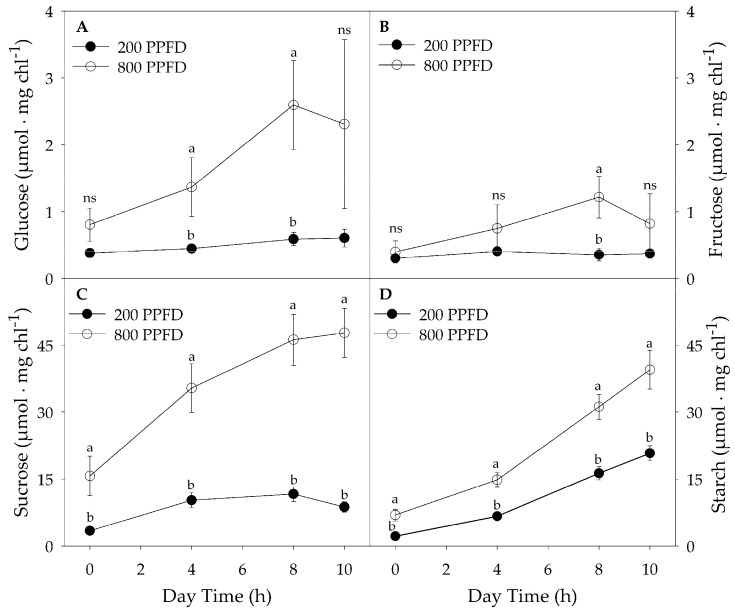
Accumulation of glucose (**A**), fructose (**B**), sucrose (**C**), and starch (**D**) during the light period in leaves of plants grown under 800 μmol m^−2^ s^−1^ (800 PPFD) and 200 μmol m^−2^ s^−1^ (200 PPFD) white light intensity. Mean values ± S.E. (n = 12). Different letters indicate significant differences at *p* = 0.05 (Fisher’s post-hoc test), ns—not significant.

**Figure 4 plants-12-00804-f004:**
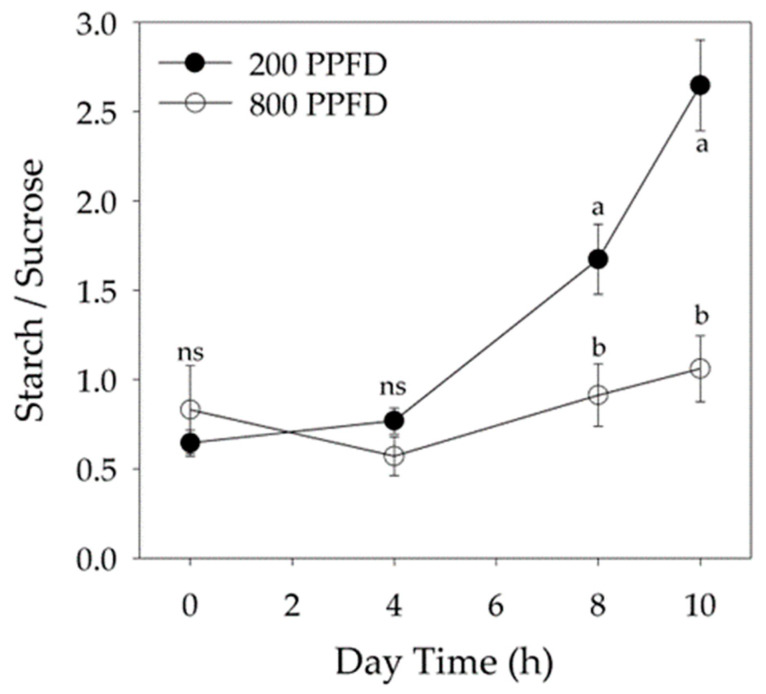
Starch to sucrose ratio during the light period in leaves of plants grown under 800 μmol m^−2^ s^−1^ (800 PPFD) and 200 μmol m^−2^ s^−1^ (200 PPFD). Mean values ± S.E. (n = 12). Different letters indicate significant differences at *p* = 0.05 (Fisher’s post-hoc test), ns—not significant.

**Figure 5 plants-12-00804-f005:**
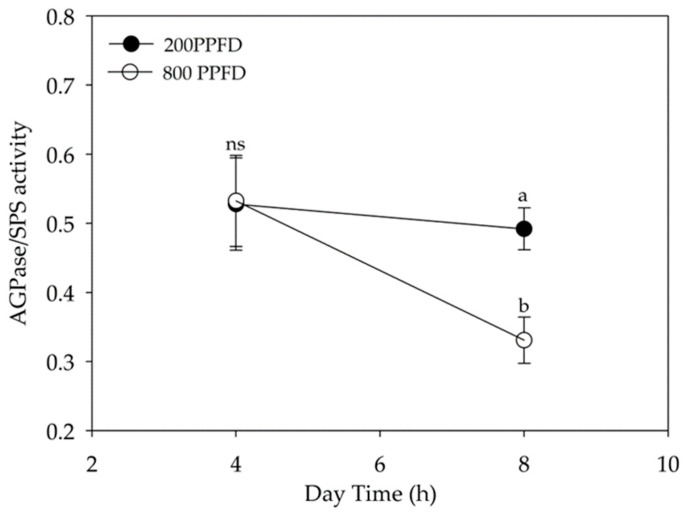
AGP/SPS ratio during the light period in spinach leaves grown in a growth chamber under 800 μmol m^−2^ s^−1^ (800 PPFD) and 200 μmol m^−2^ s^−1^ (200 PPFD) white light intensity. Mean values ± S.E. (n = 12). Different letters indicate significant differences at *p* = 0.05 (Fisher’s post-hoc test), ns—not significant.

**Table 1 plants-12-00804-t001:** Growth parameters in spinach plants grown in a growth chamber under 800 μmol m^−2^ s^−1^ (800 PPFD) and 200 μmol m^−2^ s^−1^ (200 PPFD) light intensity. Mean values ± S.E. (n = 12). Different letters indicate significant differences at *p* = 0.05 (Fisher’s post-hoc test).

	F.W. (g Plant ^−1^)	D.W. (g Plant ^−1^)	D.M. (%)
Leaf			
200 PPFD	2.78 ± 0.89 b	0.29 ± 0.03 b	10.70 ± 0.18 b
800 PPFD	7.26 ± 0.89 a	0.88 ± 0.11 a	12.00 ± 0.24 a
Root			
200 PPFD	0.88 ± 0.09 b	0.05 ± 0.01 b	6.35 ± 0.31 b
800 PPFD	3.68 ± 0.49 a	0.26 ± 0.03 a	7.35 ± 0.16 a

**Table 2 plants-12-00804-t002:** Photosynthetic parameters in leaves of spinach plants grown in a growth chamber under 800 μmol m^−2^ s^−1^ (800 PPFD) and 200 μmol m^−2^ s^−1^ (200 PPFD) white light intensity, measured at 2, 4, and 8 h of light exposure. Mean values ± S.E. (n = 12). Different letters indicate significant differences at *p* = 0.05 (Fisher’s post-hoc test).

	Time of Exposure	PPFD	
		200	800
Assimilation rate (µmol CO_2_ m^2^s^−1^)	2 h	5.0 ± 0.1 b	15.4 ± 0.4 a
	4 h	5.0 ± 0.2 b	13.3 ± 0.5 a
	8 h	4.1 ± 0.2 b	13.4 ± 0.1 a
Sub stomatal CO_2_ partial pressure (µbar bar^−1^)	2 h	360.3 ± 0.8 a	310.6 ± 2.6 b
	4 h	357.7 ± 1.9 a	319.4 ± 1.4 b
	8 h	352.6 ± 2.4 a	297.3 ± 1.8 b
Stomatal conductance (mmol CO_2_ m^2^s^−1^)	2 h	470.0 ± 15.5	517.4 ± 35.3
	4 h	460.9 ± 27.7	500.4 ± 26.5
	8 h	270.8 ± 16.8	337.7 ± 7.7

**Table 3 plants-12-00804-t003:** Starch, soluble, and total carbohydrates content (µmol C as hexose equivalent·mg^−1^ Chl) in spinach leaves grown in a growth chamber under 800 μmol m^−2^ s^−1^ (800 PPFD) and 200 μmol m^−2^ s^−1^ (200 PPFD) white light intensity, measured at the beginning of the light period (after 14 h of dark), and the end of the light period (10 h). Mean values ± S.E. (n = 12). Soluble carbohydrates are the sum of glucose, fructose, and sucrose; total carbohydrates are the sum of soluble and starch. Different letters indicate significant differences at *p* = 0.05 (Fisher’s post-hoc test), n.s.—not significant.

		Starch	Soluble Carbohydrates	Total Carbohydrates	Starch (% of Total Carbohydrate)	Soluble (% of Total Carbohydrate)
	Beginning of the day	12.9 ± 1.7 b	24.5 ± 1.9 b	37.4 ± 3.3 b	34.5 ± 2.3 n.s.	65.5 ± 2.2 n.s.
200 PPFD	End of the day	124.6 ± 10.2 b	58.3 ± 7.9 b	182.9 ± 15.6 b	68.5 ± 2.4 a	31.5 ± 2.3 b
	Variation	111.6 ± 10.9 b	33.8 ± 7.7 b	145.4 ± 16.3 b		
	Beginning of the day	41.6 ± 7.9 a	101.4 ± 28.7 a	143.0 ± 30.3 a	29.1 ± 5.8 n.s.	70.9 ± 5.8 n.s.
800 PPFD	End of the day	237.1 ± 26.1 a	305.7 ± 38.8 a	542.8 ± 35.1 a	43.7 ± 4.8 b	56.3 ± 4.8 a
	Variation	195.6 ± 28.6 a	204.3 ± 23.8 a	399.9 ± 39.0 a		

**Table 4 plants-12-00804-t004:** Estimated amount and export rates of assimilates in spinach leaves grown in a growth chamber under 800 μmol m^−2^ s^−1^ (800 PPFD) and 200 μmol m^−2^ s^−1^ (200 PPFD), based on data from Table 3. Assimilates exported during the day were calculated as the difference between the rate of photosynthesis and the carbon amount accumulated as total carbohydrates during the light period of 10 h (values shown in Table 3). Mean values ± S.E. (n = 12). Different letters indicate significant differences at *p* = 0.05 (Fisher’s post-hoc test), n.s.—not significant.

	200 PPFD	800 PPFD
Assimilation of CO_2_ (µmol C mg^−1^ chl h^−1^)	39.0 ± 3.9 b	105 ± 7.3 a
Amount of Photoassimilates (µmol C mg^−1^ chl)		
Produced during the day	390.0 ± 39.0 b	1050.0 ± 73.2 a
Accumulated during the day	146.0 ± 16.3 b	400.0 ± 39.0 a
As % on the total	37.4 ± 4.2 n.s.	38.0 ± 3.7 n.s.
Exported during the day	244.9 ± 16.3 b	650.0 ± 39.0 a
Rate of assimilates use (µmol C mg^−1^ chl h^−1^)		
During the day	24.5 ± 1.6 b	65.0 ± 3.9 a
During the night	10.4 ± 1.2 b	28.6 ± 2.8 a

**Table 5 plants-12-00804-t005:** Enzymatic activities of cFBAse, SPS_max_, and AGPase (µmol product mg^−1^ chl h^−1^) in leaves of spinach plants grown in a growth chamber under 800 µmol m^−2^ s^−1^ (800 PPFD) and 200 µmol m^−2^ s^−1^ (200 PPFD) white light intensity, and measured at 4 and 8 h of light of exposure. Mean ± S.E. (n = 12). Different letters indicate significant differences at *p* = 0.05 (Fisher’s post-hoc test).

PPFD	Time of Exposure	cFBPase	SPS_max_	AGPase
200		48.2 ± 4.94 a	21.63 ± 0.83 b	10.94 ± 0.77 b
800		63.33 ± 5.65 a	33.80 ± 2.15 a	14.20 ± 1.46 a
	4 h	56.10 ± 7.40 a	27.30 ± 2.56 a	14.27 ± 1.48 a
	8 h	52.00 ± 8.56 a	28.13 ± 3.58 a	10.85 ± 0.63 b

## Data Availability

The data presented in this study are available on request from the corresponding author (S.P.).

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
