# Peer review of "Light Intensity Affects the Assimilation Rate and Carbohydrates Partitioning in Spinach Grown in a Controlled Environment"

_plants, 2023, doi:10.3390/plants12040804_

Round 1
Reviewer 1 Report
The authors of this manuscript evaluate the photosynthetic and photoassimilate partitioning to two different light intensities (optimal, 800 umol m-2s-1, which is the higher irradiation used by the experimental setup, and suboptimal, 200 umol m-2s-1).
The work is well-designed and executed, and the manuscript is very well-written. The findings of the work show that under lower irradiances, biomass accumulation decreases, and photoassimilates are invested into ensuring maximum capture and utilization of the incoming low radiation. However, they do not detect impairments of the photosynthetic process when compared to the photosynthetic ability of spinach grown under optimal, high irradiances.
The impact of irradiance and controls over photosynthesis and assimilate partitioning have been extensively studied in the past by many directed experiments, supported by not only photosynthesis and sugar analytical data but a variety of other chlorophyll fluorescence, proteomic and metabolomic data. What I feel the authors missed to clearly outline in the introduction and discussion is:
1) the novelty of this study (they mention that their spinach was sampled at a more advanced growth stage than spinach in other studies with similar scope ) and
2) the potential practical application of their results. They do mention "vertical plant production systems" but it is unclear why vertical systems could not necessarily be accomplished with the same light input as horizontal systems unless they rely on natural light utilization and not the use of uniform output vertical light panels....
Also, the discussion is missing any inference to the differences in the rate of processes such as photorespiration at varying light intensities, competing with the efficiency of net photosynthesis, while there is evidence that photorespiration rates vary with the intensity of illumination....
I would recommend the authors consider adding the above arguments to improve the quality of their manuscript, and most importantly, highlight the novelties, otherwise, as written in its current form, it "appears" limited in its scope and may not appeal to a very wide range of audience.
Author Response
We thank the reviewer for his comments which helped us to improve the manuscript. All the requests have been addressed in the revised version of the paper. We hope our corrections satisfied the requests.

Reviewer 2 Report
Dear Collegues
I had great pleasure in reading the manuscript by S. Proietti et al "Light intensity affects the...". The authors conducted a logical study and obtained interesting evidence-based results. The data are reliable and the conclusions are valid. The article, no doubt, can be recommended for publication after explaining a number of positions.
The chosen light intensities of 200 and 800 µmol m-2 s-1 are highly questionable. The authors have published at least two articles in which they studied spinach plants at the same light intensities (https://doi.org/10.1080/14620316.2004.11511814; https://doi.org/10.1016/j.plaphy.2009.03.010).
At the same time, there are results in the literature from other researchers who have studied the effects of light of different intensities on spinach plants. The data obtained by them do not fully agree with the results presented in the reviewed manuscript.
I would like comments from the authors on three points
(1) Why do spinach plants at 200 µmol m-2 s-1 of light have such a small mass, whereas a greater biomass yield is achieved in a light region much greater than photon saturation. How can we explain such a small mass of spinach plants obtained in this work compared to the work of other authors working with the same subject;
(2) Could the authors, referring not to their own but to others' published papers, justify a light intensity of 800 µmol m-2 s-1 as optimal for spinach;
(3) Why enzyme activity is calculated on chlorophyll and not on protein, as is common practice.
Kind regards
Author Response

(The authors gave the same response as above.)

Reviewer 3 Report
Dear Authors,
Reviewer comments plants-2136421
The manuscript entitled „Light intensity affects the assimilation rate and carbohydrates partitioning in spinach grown in controlled environment“ represents a useful study aimed at a comparison of the impacts of two irradiance levels, 200 and 800 µmol m-2s-1, on photosynthesis related characteristics and assimilates accumulation in spinach. The manuscript provides valuable data providing a comparison of the two irradiance levels regarding the individual characteristics determined including basic characteristics related to biomass accumulation, net photosynthesis rate and stomatal conductance, enzymatic activities, and carbohydrate (assimilate) accumulation. The data presented in the form of tables are presented with appropriate statistical evaluation including ANOVA and Fisher´s post-hoc test while the data presented in the form of figures (graphs) have only mean and SE without an indication of statistically significant differences.
The major comment on the present manuscript is the fact that the present version contains and interpretes only the data related to the individual characteristics determined without any multidimensional statistical analyses, i.e., determination of the interrelationships between the individual characteristics to provide a more complex picture on the spinach responses to different light irradiances have to be added. I think that a complex statistical analysis of the whole set of the data determined in the study using some kind of multidimensional statistical analysis such as principal component analysis, principal coordinate analysis or cluster analysis would add more information on the interrelationships between the individual characteristics determined. I miss any summarising scheme providing a complex view on the presented data which will significantly improve the manuscript quality and provide a take-home message useful for the readers.
In addition, I have several other major comments on the present mansucript which are given below:
In Table 1 and the corresponding text, the difference between D.W. as dry weight and D.M. as dry mass has to be explained.
I do not understand the arrangement of Table 5, especially the first two columns with PPFD and time of exposure. According to the Table 5 legend, 2, 4 and 8-h time exposure should be given in the table while only 4 and 8-h time exposure is given while a 2-h time exposure is missing.
Terminology: In Abbreviations list, line 34, correct the term „stomatal conductance“ for „gs“, not „stomatic conductance.“
In Table 3 and Table 4, indication of significant differences between the presented data has to be added (ANOVA, Fisher´s post-hoc test at 0.05 level).
In Materials and methods, the source of spinach seeds cv. Gigante dÍnverno used for the experiments has to be given.
Formal comments on the text:
Abstract, line 19: Add a space between the number and the corresponding unit in „10 h light/14 h dark regime.“
Abbreviations, line 34: Correct the typing error in the compound name „fructose-6-phosphate“ (not „fructoses-6-phosphate“).
Introduction, line 61: Add a comma between the words „however“ and „it implies highly innovative…“
Results, lines 124, 127: Add a comma both preceding and following the word „respectively“ in the statements.
Table 2: Appropriate heading has to be added to the first column in Table 2, i.e., „Photosynthetic parameter“. In the column Time of exposure, a space between the number and the corresponding unit has to be added, i.e., „2 h“, „4 h“, „8 h“ instead of „2h“, „4h“, „8h“.
Line 181: Replace the word „of“ with „to“ in the statement „…prior to the dark period.“
Line 192: Modify the statement as follows: „…with respect to starch…“
Line 205: Add a comma between the words „but“ and „at the end of the day,…“
In Table 3 and Table 4, indication of significant differences between the presented data has to be added (ANOVA, Fisher´s post-hoc test at 0.05 level).
Line 230: Remove a comma in the statement: „In spinach leaves grown under low and high light intensity,…“
Line 267: Modify the word form „that“ to „than“ in the statement: „The activation state of SPS was three times higher in spinach leaves grown in high light than in those grown in low light, respectively…
Line 269: Add a comma in the statement: „…while, under 200 PPFD, the activation state was always low,…“
Discussion, line 297: Just a reference number should be given instead of the publication year in the references „Rodríguez-López et al. [24]“ and „Wu et al. [25].
Line 302: Add a comma following the words „Our data showed that in low light conditions,….“
Line 311: Replace the word „higher“ with „enhanced“ in the statement „…due to enhanced carbon assimilation and photosynthetic product availability…“
Line 325: Add a comma between the words „amount“ and „particularly“.
Line 339: Add a comma following the words „In spinach leaves,…“
Line 343: Add a comma between „44%“ and „respectively.“
Line 381: Add a comma following the words „In conclusion, our work shows that in spinach leaves,…“
Line 392: Add a comma between the words „for low and high spinach leaves“ and „respectively.“
Materials and methods, line 402: Correct the symbol for the light wavelength „λ“.
Materials and methods, line 424 and further in the text: Use „and“ instead of „e“ for „at 200 and 800 µmol m-2s-1“.
Line 453: Add a space between the number and the corresponding unit in „50 mM Hepes“.
Line 459: Replace „e“ with „and“.
Line 471: Add a comma following the words „After this treatment,…“
Conclusions, line 498: Correct the word form „production“ (not „produce“).
Line 499: Correct the word form in „Fully controlled agriculture can gain in sustainability…“ (not „full“).
Final recommendation: Reconsider after a major revision.

Author Response

(The authors gave the same response as above.)

Round 2
Reviewer 1 Report
The authors did a very good job improving their manuscript.
Author Response
We thank the reviewer for his comments, which helped us to improve our manuscript.
A number of changes have been done to improve the English language.
Reviewer 3 Report
Dear Authors,
Reviewer comments plants-2136421.R1
The revised manuscript entitled „Light intensity affects the assimilation rate and carbohydrates partitioning in spinach grown in controlled environment“ was improved by the authors in accordance with my previous comments. I especially appreciate the first paragraph in Introduction section where the main reasons leading to the present work are clearly expalined. I have only a few formal comments on the revised manuscript which, however, due to their formal nature only, can be correcetd during the proof. I can recommend the revised manuscript for publication in Plants.
Formal comments:
In Abbreviations list, I think that the authors should list and explain the abbreviation „cFBPase“ rather than „FBPase“ as „cytosolic isoform of fructose-1,6-bisphosphatase“ since the abbreviation „cFBPase“ is used in the manuscript text.
Introduction, line 139: Add the word „of“ following the word „one“ in the statement „“…in which artificial lighting represents one of the most energy requiring and expensive operations.“
Results, lines 189-195 have to be properly formatted below Table 2.
Results, line 285: Correct the statement „…while soluble carbohydrates were higher than starch at 800 PPFD (Table 3).“
Line 301: Add a comma both before and after the word „respectively“ in the statement : „In spinach leaves grown under low and high light intensity, 390 and 1050 µmol C mg-1 chl were fixed, respectively, during the 10 hours light period,…“
Line 361: Add a comma between the words „the light period“ and „respectively“.
Line 391: Add a free line between Figure 5 legend and „3. Discussion“ heading.
Discussion, line 411: Add a comma following the words „…showed that in soybean,…“
Discussion, line 632: Replace the word „with“ with „from“ in the statement „…a tendency to a reduction of cFBPase activity was observed in leaves at 200 PPFD although the difference from the leaves grown at 800 PPFD was not statistically significant.“
Line 633: Remove the publication year in the reference „Strand and co-authors [38]“.
Dicussion, line 667: Add a comma following the words „In terms of yield and quality,…“
Materials and methods, line 729: Add a space between the number and the corresponding unit in „4 h“, „6 h“ and „8 h“, respectively.
Mtaeirals and emthods, line 780: Add a space between the number and the corresponding unit in „2 mM fructose-6-phosphate“ . Correct the saccharide name „fructose-6-phosphate“ (not ůfructoses-6-phosphate).
Final recommendation: Accept after a minor revision.

Author Response
We thank again the reviewer for his comments which helped us to improve the manuscript. In addition to the suggested corrections, several changes have been done to improve the English language.
We hope our corrections satisfied the requests.